# Distributed Nodes-Based Collaborative Sustaining of Precision Clock Synchronization upon Master Clock Failure in IEEE 1588 System

**DOI:** 10.3390/s20205784

**Published:** 2020-10-13

**Authors:** Kyou Jung Son, Tae Gyu Chang

**Affiliations:** 1Department of Electrical Engineering, Myongji University, Yongin 17058, Korea; 2School of Electrical and Electronics Engineering, Chung-Ang University, Seoul 06974, Korea; tgchang@cau.ac.kr

**Keywords:** precision time synchronization, distributed sensing environment, IEEE 1588 precision time protocol

## Abstract

This paper proposes a distributed nodes-based clock synchronization method to sustain sub-microsecond precision synchronization of slave clocks upon master clock failure in IEEE 1588 PTP (precision time protocol) system. The sustaining is achieved by synchronizing the slave clocks to the estimated reference clock which is obtained from the analysis of distributed slave clocks. The proposed method consists of two clock correction functions (i.e., a self-correction and a collaborative correction, respectively). Upon master failure, the self-correction estimates a clock correction value based on the clock model which is constructed during normal PTP operation. The collaborative correction is performed in the preselected management node. The management node estimates a reference clock by collecting and analyzing clock information gathered from the other slave clocks. The performance of the proposed method is simulated by computer to show its usefulness. It is confirmed that the fifty (50) clock model-based collaborative correction maintains 10^−6^ second PTP accuracy for 10 min prolonged period after the master failure when tested with clock offset variations less than 50 ppm.

## 1. Introduction

A precision time synchronization technique is considered essential in industrial applications where various distributed data are used. In a future LAN-based distributed sensing environment in which lots of sensors participate, preciseness of time synchronization is emphasized for stable and accurate operation of systems. In general, implementation of precision time synchronization in a network environment is considered difficult since many delay features such as propagation delay, processing delay, etc., cause time synchronization error in measurement data.

The IEEE 1588 PTP (precision time protocol) standard [1] has been published for precision time synchronization (PTS) in the LAN-based network. The IEEE 1588 PTP provides sub-microsecond synchronization accuracy supported from both hardware and software to overcome the delay problem in a network environment. As a master-slave relation-based time synchronizing method, the IEEE 1588 PTP maintains its preciseness by synchronizing slave clock’s time and frequency to a master clock as shown by many previous researches [2,3,4,5]. Because of its preciseness, many time synchronization profiles based on the IEEE 1588 PTP, especially for power system applications, have been specified [6,7,8].

It is shown that the degraded time synchronizing performance directly leads to maloperation of industrial applications such as protection and control of power system apparatus [9,10,11]. Improved performance of the IEEE 1588 PTP is achieved in many ways, e.g., dealing with external disturbances and attack [11,12,13], providing enhanced clock synchronization methods [14,15], reflecting asymmetric network delay [16,17], etc. Master failure is also an important issue for reliable provision of PTS. The majority of previous research related to master failure is confined to the subject of providing backup master and preventing deterioration of PTP synchronization performance [18,19]. The direct provision of a sustaining method upon total failure of master clock without the support from backup master is eventually needed to guarantee the reliable operation of LAN-based future distributed systems. 

This paper proposes a distributed nodes-based collaborative PTS sustaining method. The proposed method is based on combined utilization of self-correction and collaborative correction. A prediction algorithm is adopted to estimate the clock drift model during the normal PTP operation period before master failure. Immediately after master failure, the self-correction prolongs PTS using the estimated clock model until a preselected management node performs collaborative correction. 

Clock data of the distributed slaves are gathered for the collaborative correction. The collaborative correction estimates a reference clock by exploring the spatially distributed clock drift characteristics (i.e., offset, transient slope and additive noise of individual clock). The estimated reference clock enables the PTS sustaining via the application of the IEEE 1588 PTP-based synchronization.

The management node is selected as a master clock through the best master clock algorithm (BMCA) [1]. Slave clock information required for the collaborative correction in the management node can be obtained through TLVs (type, length, value) defined in the IEEE 1588 PTP [1]. The TLV packet propagation delay between the management node and slave clocks must be compensated. Compensation of the packet propagation delay can be achieved by using peer-to-peer delay mechanism of the IEEE 1588 PTP applying the pre-calculated link delay from adjacent nodes. 

The performance of the proposed method was evaluated through computer simulations. It is confirmed that the 50 clock model-based collaborative correction maintains 10^−6^ second PTP accuracy for a 10 min prolonged period after master failure when tested with clock offset variations less than 50 ppm. The operability and the performance of the proposed method are also validated from an embedded DSP hardware operation test. The operation test showed that the proposed algorithm sustains sub-microsecond precision for 124 s after master failure. This is considered very long compared with 20 s which is measured when the proposed algorithm is not applied. In contrast to other redundant equipment-based backup solutions such as backup master, high-availability seamless redundancy (HSR) [20], parallel redundancy protocol (PRP) [20] etc., the proposed algorithm is considered a software-based solution which can provide PTS sustaining means upon the total failure of master clock and backup equipment. Therefore, the proposed PTS sustaining method is expected to provide a significant contribution to the reliable operation of industrial applications.

## 2. Distributed Nodes-Based Collaborative PTS Sustaining Method

In this section, basic principles of the proposed distributed nodes-based collaborative PTS sustaining method are described. The conceptual diagram of the proposed method is described in Figure 1. Temporal and spatial data of distributed nodes are used to compensate clock disturbances and eventually provide collaboration-based reference clock information. The application of linear prediction algorithms to the temporal data can provide estimation results of clock frequency, which is very useful to maintain precision synchronization during the clock self-correction period. Ensemble averaging is a good candidate for the effective exploration of the spatially distributed clock drift characteristics (i.e., offset, transient slope and the additive noise of individual clocks) providing accurate reference clock estimation by suppressing clock offsets and disturbances. The collection of the slave clocks’ data can be achieved by using management functions defined in the IEEE 1588 PTP standard [1]. Multirate sampled data is used to compensate low-rate disturbances which are distinguished from clock’s statistical features. 

The operation process of the proposed method is illustrated in Figure 2. The proposed method is divided into two functions: the clock self-correction and collaborative clock correction. A prediction algorithm is adopted to estimate the clock drift model during the normal PTP operation period. Immediately after master failure, the self-correction prolongs PTS using the estimated clock model until a preselected management node performs collaborative correction. The collaborative clock correction algorithm starts after the activation of the management node. The management node is activated when it becomes a master clock by the result of the BMCA of the IEEE 1588 PTP [1]. The initialization process of the collaborative correction algorithm includes adaptive configuration of PTP parameters and sustaining PTP specifications including precision class and sustaining duration requirements. The management node collects the spatiotemporal data from distributed slave clocks to estimate the collaboration-based reference clock. After the estimation, the management node and slave clocks synchronize their clocks to the reference clock, thus PTS sustaining is achieved.

### 2.1. Clock Self-Correction 

The proposed clock self-correction constructs a slave clock’s own clock model by using clock compensation data obtained from the PTP-based synchronizing operation. As shown in Equation (1), the constructed clock model reflects external disturbances and clock’s characteristics such as offset and drift. After master failure, the clock model is used to maintain PTS by estimating a clock self-correction value.
(1)$$F_{n}\left(k\right)=F_{ref}+D_{n}\left(k\right)+\upsilon_{n}\left(k\right),$$
where k denotes the discrete time index which is updated with synchronization period $T_{sync}$. $F_{n}\left(k\right)$ denotes the frequency of n th slave clock. $F_{ref}$ denotes the reference frequency. $D_{n}\left(k\right)$ denotes the frequency deviation parameter which contains both offset and drift. $\upsilon_{n}\left(k\right)$ denotes the additive disturbance.

It can be assumed that clock frequency does not have much variations resulting in a quasi-stationary characteristic, which enables convergence of adaptive filtering algorithms under a controlled environment preventing external airflow and keeping minimized variations on temperature and humidity. Therefore, the frequency deviation parameter $D_{n}\left(k\right)$ can be estimated with its past information as shown in Equation (2). After master failure, the self-correction algorithm provides the estimated frequency deviation value, ${\hat{D}}_{n}\left(k\right)$, to compensate $D_{n}\left(k\right)$.
(2)$${\hat{D}}_{n}\left(k+1\right)=D_{n}\left(k\right)w_{n}^{1}\left(k\right)+D_{n}\left(k-1\right)w_{n}^{2}\left(k\right)\cdots \;+D_{n}\left(k-\left(M-2\right)\right)w_{n}^{M-1}\left(k\right),$$
where $w_{n}^{l}\left(k\right)$ denotes l th prediction-filter coefficient of n th slave clock. M denotes the number of filter coefficients. The filter coefficients are updated during the normal PTP operation. 

The flow chart of the self-correction is shown in Figure 3. During the periodical PTP-based synchronizing operation, a slave clock updates its clock model by using clock correction values obtained from the result of the synchronizing process with a master clock. The clock model update process is repeated until the occasion of master failure. After master failure, the clock self-correction is activated instead of the master-slave relation-based synchronizing process. The self-correction value is estimated from the constructed clock model. The slave clock synchronizes its clock parameter to the estimated self-correction value. The clock model must not be updated after the master failure since there is no reference information.

The performance of the clock self-correction mainly depends on the accuracy of the constructed clock model. The main target of the modeling is estimating oscillator’s characteristics such as offset and drift. Unless the surrounding environment of PTP equipment is abruptly varying, the oscillator’s characteristics can be estimated by application of adaptive modeling techniques to the clock correction values obtained from the PTP process. Figure 4 is an example of the clock modeling process. The adaptive filter update block adjusts the clock model as the new clock correction value is periodically calculated.

The LMS algorithm is considered a proper choice for the slowly changing clock model in view of convergence and accuracy performance as shown in Equation (3).
(3)$$w_{n}\left(k+1\right)\;=\;w_{n}\left(k\right)+\mu \times D_{n}\left(k\right)\times J_{n}\left(k\right),$$
where μ denotes a step-size parameter. $w_{n}\left(k\right)$ denotes the prediction-filter coefficient vector of *n* th slave clock. $D_{n}\left(k\right)$ denotes the vector of the frequency deviation parameter of *n* th slave clock. $J_{n}\left(k\right)$ denotes the frequency prediction error of *n* th slave clock. In general, the application of other optimum algorithms will show comparable or better performance and benefit from the utilization of more data and stochastic information than the utilization of instantaneous data in the LMS algorithm. However, with the clock used in this research, it is shown that the clock frequency does not have much variations resulting in a quasi-stationary characteristic over several minutes by providing proper control of the clock operating environment. 

### 2.2. Collaborative Clock Correction Algorithm

To maximize prolonged PTS, the proposed collaborative clock correction must be performed based on the results of the self-correction. Reference clock estimation by exploring the spatially distributed clock drift characteristics of an individual clock is a key element constructing the collaborative correction. Ensemble averaging of spatially distributed clock characteristics is adopted in the implementation of the collaborative correction to suppress the effect of clock offsets and disturbances as shown in Equations (4)–(6).
(4)$$Ensemble\;average\;of\;clocks=\frac{1}{N}\sum_{n=1}^{N}t_{n}\left(k\right);$$
(5)$$t_{n}\left(k\right)=t_{n}\left(k-1\right)+T_{sync}+\;E_{n}\left(k\right);$$
(6)$$E_{n}\left(k\right)=\frac{F_{n}\left(k-1\right)-F_{ref}}{F_{ref}}\times T_{sync},$$
where, $t_{n}\left(k\right)$ denotes the measured time at time index *k* of *n* th slave clock. $T_{sync}$ denotes the PTP time synchronization period. $E_{n}\left(k\right)$ denotes the time error measured at time index *k*. Effectiveness of applying the ensemble averaging with Equations (4)–(6) is supported considering that the behaviors of widely spread clocks tend to show increased independence.

The flow chart of the proposed collaborative clock correction is shown in Figure 5. After master failure, the management node gathers the clock data of slave clocks through the data sharing process. TLV packets, which are used to get or set the PTP clock’s parameters as defined in the IEEE 1588 PTP, are used to gather the clock information. Ensemble average of the gathered clock information provides the estimated collaboration-based reference clock. The reference clock is used to synchronize all the slave clocks via the IEEE 1588 PTP-based synchronization. 

Propagation delay occurring in the collection of the clock information must be compensated to enhance the preciseness of the generated reference time. Under a symmetric network condition, propagation delay compensation can be achieved by using peer-to-peer delay mechanism of the IEEE 1588 PTP. Every node can compensate propagation delay from adjacent nodes without requiring additional processes since they know link delay occurs against their adjacent nodes in the PTP environment where peer-to-peer delay mechanism is adopted. A priori characterization of end-to-end propagation delay, which requires additional execution of the end-to-end delay measurement process of the IEEE 1588 PTP, can also be utilized in the case of a heterogeneous PTP network environment.

The two functions of the proposed method (i.e., the clock self-correction and the collaborative clock correction) should be performed together to gain the maximized sustaining ability of PTS. It is especially important that the clock self-correction process be performed as soon as possible after the master failure. This is because sub-microsecond preciseness of clock is only maintained for a very short time without reference information.

## 3. Simulations

In this section, performance of the proposed PTS sustaining method was evaluated with computer simulations. The operation of the proposed method is also tested with an embedded DSP hardware system. To verify the excellence of the proposed method, time synchronization performance was compared with a PTP slave clock which only performs the original PTP process.

### 3.1. Performance Evaluation with Computer Simulations

Performances of the proposed PTS sustaining algorithm were evaluated with a Matlab program. The tab number and a step-size of the LMS algorithm was set to 8 and 0.005, respectively. The time synchronization period was set to 1 s. It is assumed that the propagation delay that occurred during the clock data sharing is compensated for.

The clock model used in the simulation was created by using a 2-tab IIR filter, where the filter input was a randomly chosen offset value to reflect a saturation characteristic. The nominal frequency of the clock was set to 20 MHz. A zero-mean white gaussian noise with a standard deviation of 1 Hz was added as a disturbance. Two parameters of a clock servo algorithm (i.e., a proportional and an integral constant) were set to 0.1 and 0.3, respectively. Examples of the clock models are shown in Figure 6. 

Time synchronization performances of PTP clocks were observed during the simulation of which the duration was 5000 s. The normal PTP operation was performed for the first half of the duration (i.e., 2500 s). At 2501 s, the self-correction algorithm was activated because of the master failure. The collaborative correction algorithm operated after 20 s of the self-correction algorithm. 

The time synchronization performances of three PTP slave clocks (i.e., a management node with the proposed PTS sustaining algorithm, a self-correction only applied slave clock and a normal slave clock without the proposed method) were compared. Fifty clocks including the management node participated in the collaborative correction. All 50 clocks synchronized to the same master during the normal PTP operation. After the master failure, the management node became a new master and the remaining 49 clocks synchronized to the management node.

The results of the performance comparison are shown in Figure 7. Before the master failure, the PTP clocks maintained sub-microsecond synchronizing performance by the normal PTP operation. The effectiveness of the proposed PTS sustaining algorithm is well shown by comparing the slopes of the time synchronization error of the PTP clocks after the master failure. The PTP clock with the proposed algorithm has the lowest time synchronization error slope. On the other hand, the time synchronization error of the PTP clock without the proposed algorithm drastically increased so its sub-microsecond synchronization performance was sustained for a relatively short time. The time synchronization error of the self-correction only applied PTP clock increased as the model error between the estimated clock and the real clock increased. The collaborative correction must substitute the self-correction algorithm before the model error results in large time error. 

Percent duration of PTS sustaining was observed for varying standard deviation of clock offset to evaluate PTS sustaining performance of the proposed algorithm. Target sustaining duration was set to 10 min. The simulation was performed 500 times and the percent duration was obtained by averaging the simulation results. Fifty clocks participated in estimating the collaboration-based reference clock. The results of the simulation are summarized in Table 1. It is confirmed that the performance of the PTP clock with the proposed algorithm surpasses the performance of the normal PTP clock. 

### 3.2. Operation Test on Embedded DSP System

The proposed algorithm was implemented with TMDSIDK572 multi-processor boards to illustrate the operation of the proposed method. The oscillator of the board is a normal crystal oscillator which has 30 ppm of clock accuracy at 20 MHz. The Linuxptp reference source code [21] was used to enable basic PTP functions. 

Two EVM boards and PTP equipment were used for the self-correction operation test. The two EVM boards behaved as slave clocks during the normal PTP operation. One of the two EVM boards was preselected as a management node. A LANTIME M600 high end time server was used as a grandmaster clock and an RSG 2488 PTP supported network switch was used as a transparent clock. 1-PPS (pulse per second) signals of the EVM boards and the grandmaster clock were used to compare PTS sustaining performance (i.e., maintaining sub-microsecond precision). The master failure was simulated by pulling out the connected ethernet cable.

The observation results of 1-PPS signals during the master failure are shown in Figure 8. Three 1-PPS signals (i.e., the management node, the slave clock without the proposed algorithm and the grandmaster clock) were captured by an oscilloscope. The 1-PPS signal of the master clock was used as a reference signal in comparing PTS sustaining performances. When the master failure occurred, both the management node and the slave clock maintained sub-microsecond precision as shown in Figure 8a. The 1-PPS signal of the slave clock exceeds a 1 microsecond time difference after 20 s. On the other hand, the management node sustains a 1 microsecond time difference for 124 s which is very long time compared with the slave clock’s performance, as shown in Figure 8b.

The proposed collaborative PTS sustaining algorithm was implemented with three EVM boards and the PTP equipment to confirm the operation of the proposed algorithm including the collaborative correction phase. Because of difficulty accessing the commercially provided PTP equipment, propagation delay compensation is indirectly reflected by pre-calculated delay values among the EVM boards. The example of PTS sustaining obtained from the experimental test system is illustrated in Figure 9. It is observed that the collaborative correction sustains more than 1 min which is expected to prolong as the number of slave clocks increases. It is worth mentioning that major implementation elements of the proposed PTS sustaining algorithm (i.e., the timestamping and delay correction of the management packets) are readily realizable with the availability of directly modifiable PTP equipment. 

## 4. Conclusions

This paper proposes a PTS sustaining method which prolongs sub-microsecond time synchronization precision upon master failure. PTS sustaining is achieved by the successive application of a self-correction and a collaborative correction algorithm. The self-correction performs the prediction of clock drift characteristics and the collaborative correction gathers clock information of slave nodes and estimates a reference clock for the management node-based PTP synchronization. The proposed method tries to maximize PTS sustaining after master failure by using slave clocks having relatively low clock quality. The computer simulation results verified that PTS sustaining performance of the proposed algorithm applied to a PTP clock surpasses the performance of a normal slave clock. The operability of the proposed algorithm was validated from the embedded DSP hardware operation test. The operation test also showed that the proposed algorithm sustains sub-microsecond precision for 124 s after master failure. This is considered a very long time compared with the PTS sustaining performance of the normal slave clock, which is 20 s. The proposed algorithm is considered a software-based solution which can provide PTS sustaining means upon the total failure of a master clock and backup equipment. Therefore, the proposed PTS sustaining method is expected to provide a significant contribution for the reliable operation of industrial applications.

Proper clock grouping can provide enhanced accuracy in the estimation of a reference clock. The design of the clock grouping, which is left to future research, requires in-depth studies on various system-related factors including sustaining PTP specifications, network topology and traffic, system configuration and clock behavior modeling.

## Figures and Tables

**Figure 1 sensors-20-05784-f001:**
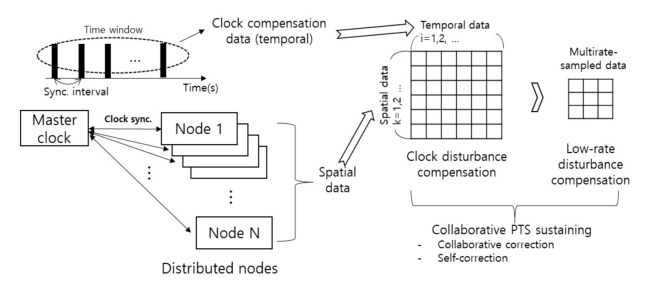
Conceptual diagram of the proposed precision time synchronization (PTS) method.

**Figure 2 sensors-20-05784-f002:**
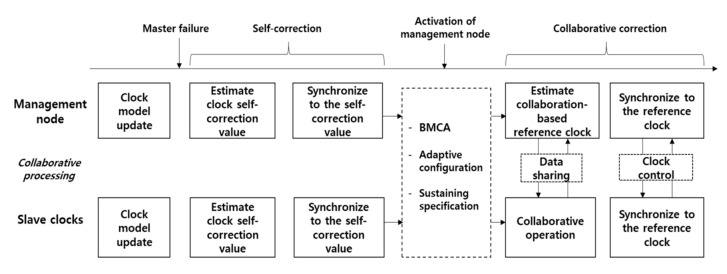
Operation process of the proposed method.

**Figure 3 sensors-20-05784-f003:**
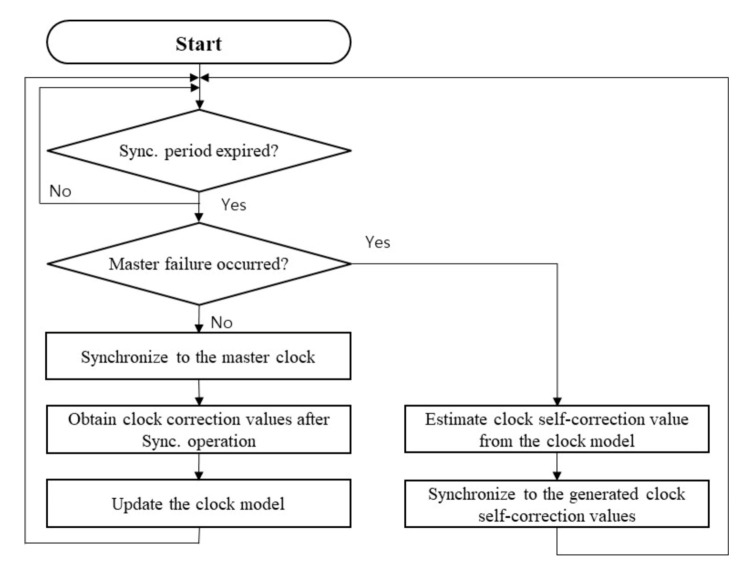
The flow chart of the proposed clock rate self-correction algorithm.

**Figure 4 sensors-20-05784-f004:**
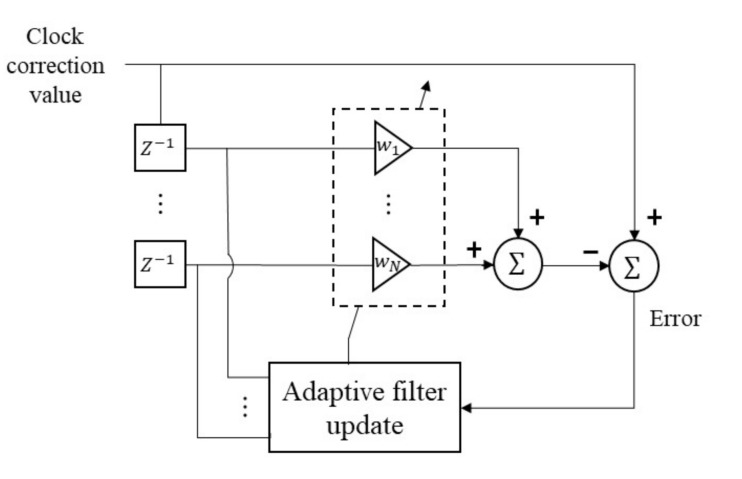
An example of the clock modeling process.

**Figure 5 sensors-20-05784-f005:**
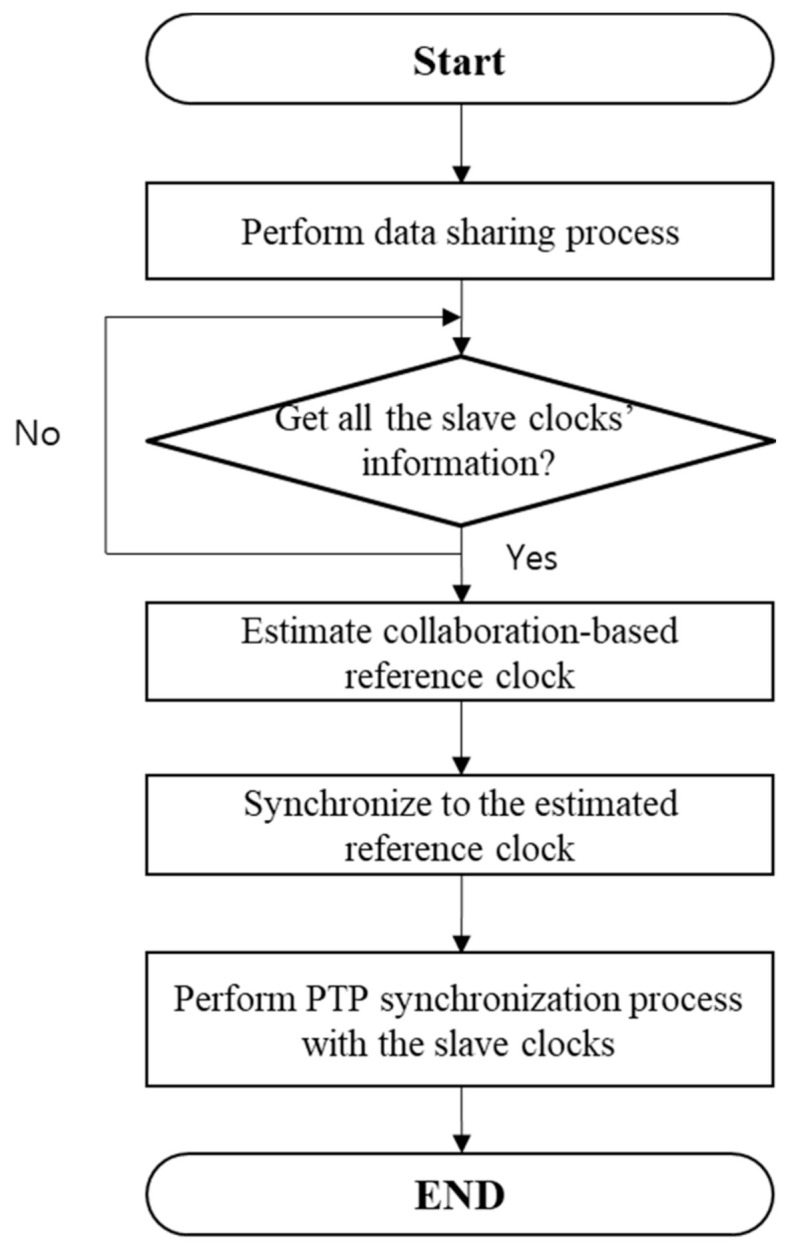
The flow chart of the proposed collaborative clock correction process.

**Figure 6 sensors-20-05784-f006:**
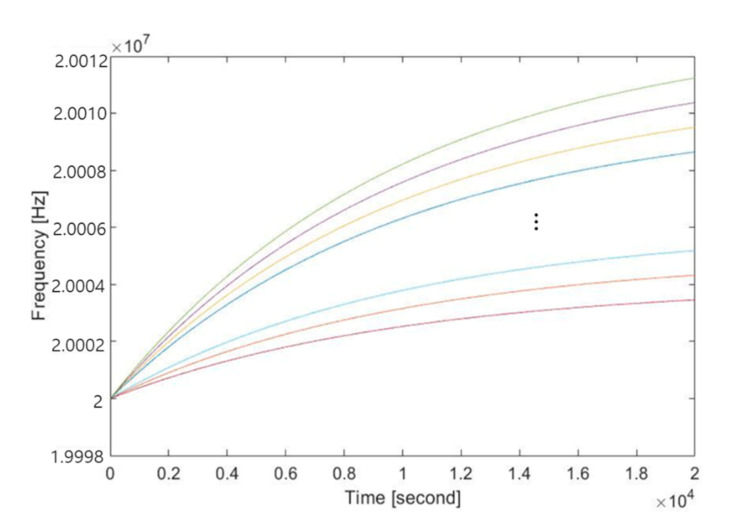
An example of the clock models used in the simulation.

**Figure 7 sensors-20-05784-f007:**
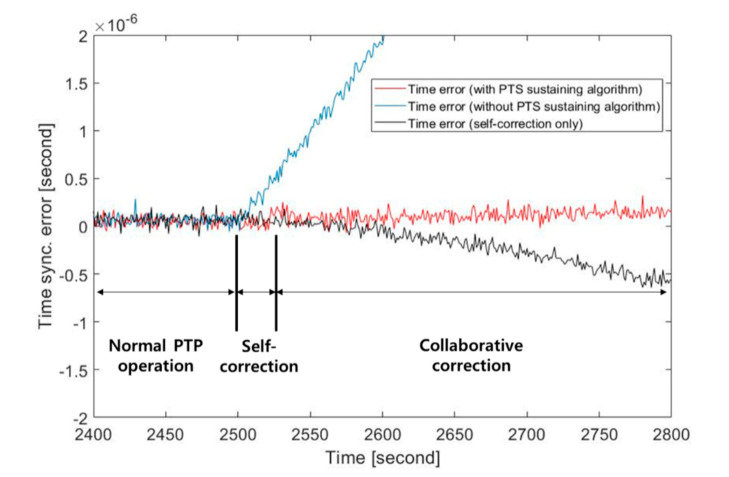
Comparison of time synchronization performance of precision time protocol (PTP) clocks.

**Figure 8 sensors-20-05784-f008:**
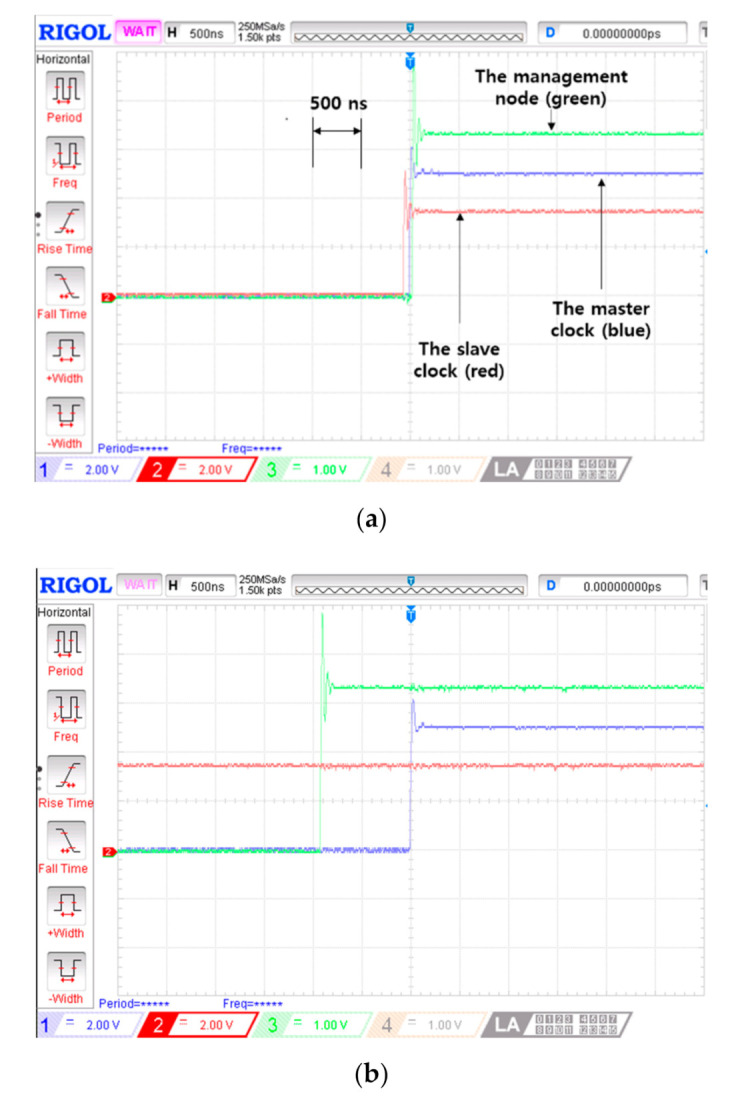
Operation test result of the proposed self-correction. Three 1 pulse per second(PPS) signal comparisons over time after the master failure. (**a**) 0 s (at the time of the master failure occurrence); and (**b**) 124 s.

**Figure 9 sensors-20-05784-f009:**
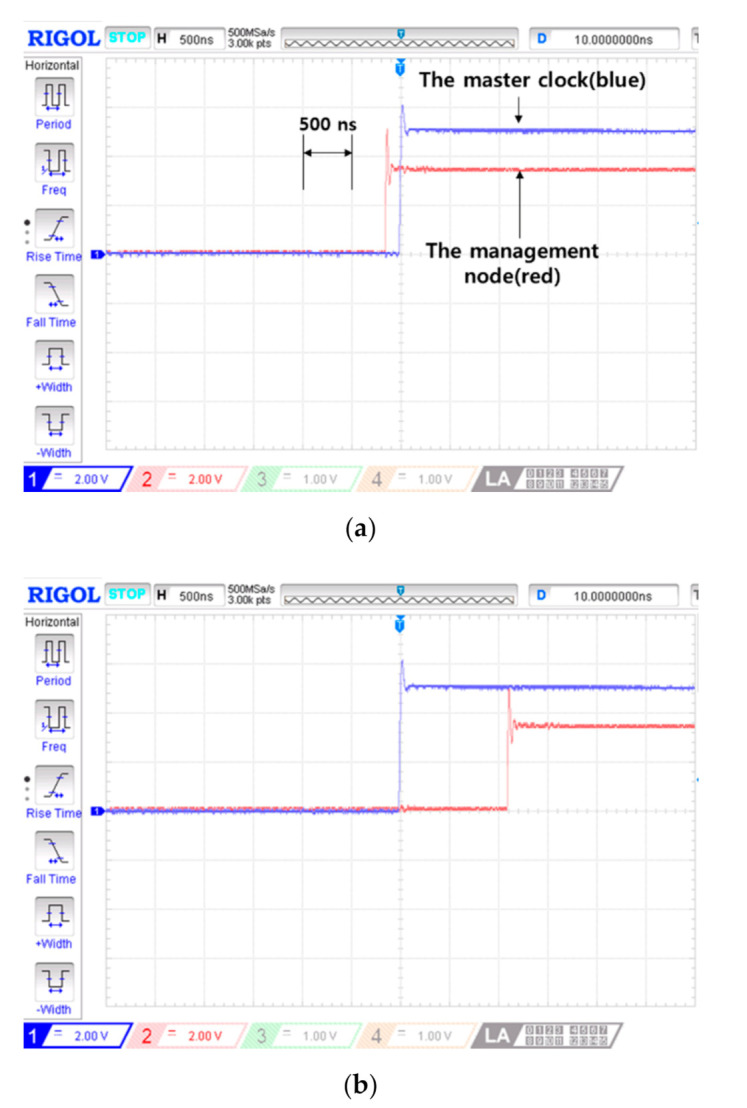
Operation test result of the proposed collaborative PTS sustaining algorithm. Two 1-PPS signal comparisons over time after the master failure. (**a**) 0 s (at the time of the master failure occurrence); and (**b**) 62 s.

**Table 1 sensors-20-05784-t001:** Comparison of percent duration of PTS sustaining of PTP clocks.

Standard Deviation of Clock Offset [kHz]	Percent Duration of PTS Sustaining [%]	
	Without PTS Sustaining Algorithm	With PTS Sustaining Algorithm
1	62.05	100.00
5	20.81	77.82
7	14.82	69.10
10	10.32	57.19
15	6.17	41.63
20	5.53	28.76

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
