# Peer review of "Distributed Nodes-Based Collaborative Sustaining of Precision Clock Synchronization upon Master Clock Failure in IEEE 1588 System"

_sensors, 2020, doi:10.3390/s20205784_

Round 1
Reviewer 1 Report
This paper deals with retaining accurate synchronization in a PTP network as long as possible after the master clock becomes unavailable. It uses a self-correction algorithm for each slave node and a collaborative clock synchronization algorithm to maintain a common global time reference.
The description of the methods and algorithms is repeated at somewhat the same (low) level of detail several times, which is repetitive and not useful.
line 139: " Unless the surrounding environment of PTP equipment is abruptly varying, the oscillator’s characteristics can be estimated by the application of adaptive modeling techniques to the clock correction values obtained from the PTP process." The authors should motivate that this is readily achievable in practice. Equipment self-heating, airflow, etc. can easily change the oscillator behavior. Is a normal crystal oscillator in a proper enclosure or a TCXO enough for this to work as described or are more accurate (e.g. ovenized) oscillators required to achieve the described performance? Authors should give some idea of the equipment involved/required. The hardware used is mentioned later in the paper, but its key characteristics are barely described.
line 169: " The collaboration-based reference clock can be estimated
from the collected spatio-temporal time data by the application of signal processing techniques. The simplest example of estimating the reference time is conducting ensemble average for all the collected data which is very effective in suppressing a noise component. " Which signal processing techniques in particular? And these claims must have citations.
line 200: " The performances of the proposed PTS sustaining algorithm were evaluated with Matlab program. An adaptive LMS (Least Mean Square) algorithm is used as a self-clock model estimator in realizing the self-correction algorithm." I think I need more details to reproduce these results. For one the exact LMS algorithm and the clock model and parameters are needed. Is it a simple two-state model?
line 252: " Because of the difficulty in realization of PTP network with lots of slave clocks, the collaborative correction algorithm was set not to be activated." What difficulty? Is it too expensive?
The simulation assumes that the "propagation delay occurred during clock data sharing is compensated". How do the practical experiments deal with this?
Overall, this paper has merit. I am not extremely familiar with the PTP research branch within the overall clock synchronization topic, but this work looks sufficiently novel to me.
However, the methods and algorithms are described in insufficient detail to allow for reproducing these results reliably. This needs to be addressed before this paper is published. In contrast, the overview (low detail) description of the methods and algorithms feels repetitive, which is distracting while reading this work.
What also stands out is that no comparison with other similar/alternative solutions to this problem are made, nor are alternatives cited or even described; there is no "Related Work" section (or even a paragraph in the introduction). Only algorithms that improve PTP while it is still live are referenced in the introduction; nothing that deals with master clock outage. I doubt there is no other work done in retaining synchronization after master clock failure (for PTP or other similar domains). Authors must survey for other solutions or at least clearly state that they found none.
Finally, this work is mostly based on simulation. The practical experiments use only a small number of nodes and the collaborative part of the proposed method is not tested there (as far as I understood). This, together with the lack of comparison with other solutions, limits the merit of this work. The reasons for the lack of extensive practical testing are left unclear.
Author Response
We thank the reviewer for the considerations on our submitted manuscript. We have revised the manuscript reflecting all the reviewer's requirements and comments. The changes we have made in the revised manuscript are summarized in the attached file.

Reviewer 2 Report
The authors of this paper propose a collaborative Precision Time Synchronization (PTS) mechanism intended for sub-microsecond synchronization precision when master clock failures occur in an IEEE 1588 PTP (Precision Time Protocol) system. The proposal achieves PTS by synchronizing the estimation of reference clocks, which is obtained from the collaboration of distributed slave clocks. The proposal comprises two correction mechanisms: a self-correction and a collaborative correction. Then, the algorithms are tested by means of computer simulations as well as with DSP hardware. The results show that the proposal outperforms a normal slave clock with Matlab computer simulations, and also it sustains sub-microsecond precision for 124 seconds after master failure.
The paper is well structured and flows well. However, it must be revised for English grammar and typos. There are some points that should be corrected. For example, starting with the abstract, line 11 says "This paper proposed a distributed ...", which sounds weird; it could rather say "This paper proposes a distributed ...". Lines 16-17 say "The self-correction algorithm, which is firstly operating algorithm upon master failure ..." is badly written. Details like these are present throughout all the paper.
Regarding the evaluation by means of computer simulation, the self-correction algorithm uses an adaptive LMS (Least Mean Square) filter as a self-clock model estimator. Moreover, the authors assume that the propagation delay that occurs during data sharing is compensated. These are key choices in the proposal; thus, it would be desirable that the authors provide additional descriptions about them. For example, how about choosing a couple more different filters for estimation? How much time would consume the compensation of the propagation delay? What methods would be adequate for such compensation? Also, choosing 50 clocks for the collaborative correction algorithm is an important choice. Therefore, how is this implemented in Matlab? And, how this would impact the performance on hardware implementation?
Finally, it is mentioned that the characteristics of the slave clocks are very important. Such clocks are divided into groups and such a grouping impacts the performance of the proposal. Thus, additional comments would be desirable in the contribution rather than just mentioning them in the Conclusions section.
Author Response

(The authors gave the same response as above.)

Reviewer 3 Report
The used algorithms are not detailed enough even to consider the reproduction of results, as their description is vague. For example, the details of the LMS adaptive filter update, applied constants, etc. should be presented. Actually, the collaborative part of the algorithm is introduced in even less detail, we do not know how the actual corrections worked out based on the measurements. We should see equations describing the algorithms, not some cryptic textual descriptions and flow charts (they are needed, but they are not enough).
How it is assured, for example, that the LMS is convergent to the real system? The simulation and the presented working example may show some "lucky cases", in which it is, but in real life, it may not for a large number of actual systems.
The simulations are inadequate, as a non-realistic clock model is used in my view (it is very simple), and typical failure scenarios are not investigated. For example, the disappearance of the master is a random process relative to the actual state of the system, therefore, detailed statistical analyses should be presented. The paper mentions that it tested/simulated 50 clocks, but it is not presented how these clocks were operated, and why it is possible to substitute testing of 50 clocks with one single clock with 50 different instances of random clock behavior. We do not know if Fig. 7. is the worst-case, just a case (random selection), or best case, but it is known that is a single case (not statistical analyses).
The real measurements only check the first phase of the algorithm, therefore, they do not prove all claims of the paper. The whole algorithm should be implemented, tested, and the results presented in the paper.
I do not see how the solution operated in case of a Byzantine error assumption (which is the case for safety-critical systems, for which the paper targets its results). BMCA is inappropriate in such a scenario. How the proposed algorithm may operate in redundant systems (dual or triple modular redundancy)?
The paper needs a thorough English review and correction, especially punctuation lacks.
Author Response

(The authors gave the same response as above.)

Round 2
Reviewer 1 Report
OK, I think all my comments and concerns about the content of the paper were addressed appropriately.
Only, the quality of the English text did not improve. In fact, some of the additions contain new problematic sentences.
Author Response
We really appreciate your considerations on our manuscript. To reflect the reviewer’s comment, we revised the manuscript with correction of English.
Reviewer 2 Report
The corrections made have greatly improved the contribution. Moreover, the added explanations about the estimator used as well as those for the propagation delay compensation provide a most complete explanation of the contribution for the reader.
Author Response
We really appreciate your considerations on our manuscript.
Reviewer 3 Report
I think the prove the claims of the paper some real measurements to prove the applicability of the second phase of the algorithm. If not with 50 nodes, at least with several (4-6) nodes. Without that, we do not know if the hidden complexity of the real system makes it impossible to reach the claimed performance.
Other than that, the paper can be accepted in my view...
Author Response
We really appreciate your considerations on our manuscript. The changes we have made in the revised manuscript are illustrated in the attached file.
